# OpenReview forum: "Wisdom is Knowing What not to Say: Hallucination-Free LLMs Unlearning via Attention Shifting"
_NeurIPS.cc/2025/Conference — NeurIPS 2025 poster_

### Official Review · Reviewer_Tnjq · 2025-05-31

**Clarity:** 4
**Significance:** 3
**Originality:** 3
**Rating:** 4
**Confidence:** 4

**Summary:**

This paper proposes a novel framework called Attention Shifting (AS) for machine unlearning in Large Language Models (LLMs). AS aims to remove specific knowledge from a model while minimizing performance degradation and hallucination. Unlike previous aggressive (e.g., GA, NPO) or conservative (e.g., ULD, IHL) unlearning methods, AS selectively suppresses attention to fact-bearing tokens in the unlearning dataset and reinforces attention to semantically important tokens in retained data. This is accomplished using adapter-based attention modulation and a dual-loss formulation. AS is evaluated on two benchmarks, ToFU and TDEC, demonstrating superior trade-offs in preserving model utility and minimizing hallucinated responses compared to state-of-the-art baselines.

**Questions:**

1. Could the authors elaborate on how robust the attention-based suppression is to adversarial probing techniques that try to extract memorized content through indirect queries? This would help assess the security guarantees of the method.
2. The paper mentions that latent traces may still reside in internal representations (Section 4.4). Could the authors comment on potential extensions that might combine AS with representation-level erasure techniques?
3. AS performs well under limited retention data conditions. How does it scale when applied to very large-scale models (e.g., GPT-3.5 or GPT-4 size)? Are there computational constraints?
4. The adapter placement strategy is described in Appendix B.1. Did the authors explore the relative benefits of inserting adapters after query/key projections versus the value projection in more detail? A small ablation could help clarify this design choice.
5. Could the authors provide more real-world examples (e.g., privacy-sensitive medical data) to illustrate AS’s applicability outside synthetic benchmarks like ToFU?

**Ethical Concerns:**

["NO or VERY MINOR ethics concerns only"]

**Final Justification:**

Rebuttal clarified some details I was looking for. I've updated my rating.

**Limitations:**

The paper has a dedicated limitations section (Sec 4.4) and responsibly discusses dataset quality dependency, incomplete internal erasure, and trade-offs in generative applications.

**Paper Formatting Concerns:**

None.

**Quality:**

4

**Strengths And Weaknesses:**

**Strengths**:
- The paper is well-executed with a comprehensive methodological design, clear theoretical justification (Appendix A), and strong empirical validation. The use of both ToFU and TDEC benchmarks demonstrates robustness across diverse data types.
- The paper is generally clear and well-organized, especially in describing the motivation for AS and its implementation through attention modulation. However, some parts, such as Sec 3.3 on knowledge maintenance and Appendix B.1 on adapter architecture, could be more accessible with illustrative diagrams or concise summaries.
- The work addresses an increasingly critical issue in LLM deployment—privacy-preserving unlearning without hallucination. The AS framework has practical implications for responsible AI, particularly in GDPR-compliant environments and high-stakes domains such as healthcare or law.
- The paper introduces an original concept of attention-based unlearning that operates at a different level than prior logits-based or loss-based approaches. The idea of enforcing hallucination-resistance through attention suppression and reinforcement is novel and impactful.
- AS achieves significant performance retention in general and neighboring knowledge, while maintaining high unlearning effectiveness (e.g., up to 15% and 10% improvements on ToFU and TDEC benchmarks respectively).
- The hallucination suppression results (0% hallucination and reproduction rates) are compelling and further distinguish the method from competitors like ULD and IHL.
- The unlearning evaluation strategy is thorough, involving multiple metrics (ROUGE-L, TR, FQ, hallucination rate, reproduction rate), model sizes (GPT-NEO variants), and progressive forgetting scenarios.

**Weaknesses**:
- A major limitation is addressing the interpretability of attention shifts in more intuitive terms, additional ablations on adapter placement, or demonstrating AS’s efficacy in real-world or multilingual contexts.
- A noted limitation is that AS focuses on behavioral unlearning and may not guarantee complete representational erasure. This is acknowledged in Sec 4.4 but remains a relevant caveat for high-security applications.
- The effectiveness of AS is reduced on noisier datasets like TDEC compared to structured ones like ToFU, suggesting sensitivity to input quality for token importance estimation.

---

> ### Author Rebuttal · Authors · 2025-07-29
>
> Thank you for highlighting the novelty of our attention-based unlearning approach, its strong empirical performance across ToFU and TDEC, and its practical value for privacy-preserving applications. We also appreciate your recognition of the method’s clarity, robustness across settings, and zero-hallucination results. Please find our response to your question below.
>
> ---
> >**[Q1]** Interpretability attention shifts & Real-World Gaps
> ---
> **A1:** We agree that a more intuitive understanding of how attention is reallocated would enhance interpretability.
> 1. We plan to include additional visualizations of attention heatmaps (e.g., before and after suppression) and their correlation with model uncertainty. In particular, we aim to show how the redirection of attention to neutral tokens increases entropy and reduces model confidence on forgotten facts, clarifying how AS behaviorally enforces unlearning.
>
> 2. We appreciate the suggestion to further explore the applicability of our method in real-world and multilingual settings. While our current evaluation focuses on two established unlearning benchmarks, ToFU and TDEC, these datasets are intentionally chosen to cover a broad spectrum of unlearning scenarios. ToFU provides a controlled, structured synthetic setup that enables precise measurement of unlearning behaviors, while TDEC contains noisy, real-world factual triples derived from natural corpora, offering more realistic challenges. The fact that our method performs consistently well across both datasets, and across two model architectures (LLaMA and GPT), demonstrates its adaptability and robustness.
>
> 3. We agree that multilingual unlearning remains an open and important direction. However, to the best of our knowledge, there are currently no standardized multilingual benchmarks specifically designed for selective unlearning evaluation. We see this as an exciting area for future work, and we plan to explore multilingual extensions of our attention-shifting mechanism, including cross-lingual unlearning and language-specific retention strategies.
>
> ---
> >**[Q2]** Did the authors evaluate the impact of placing adapters after query/key vs. value projections?
> ---
> **A2:** Thank you for the thoughtful question. We acknowledge that adapter placement is a key design choice. In Appendix B.1, we discuss that placing adapters after Q/K projections allows indirect control of attention weights with minimal overhead. Additionally, we have experimented with placing adapters after the value (V) projection, and find it provides comparable unlearning performance while remaining lightweight.
>
> Furthermore, we have included quantitative comparisons of adapter placement in terms of memory overhead, parameter update size, and unlearning effectiveness. We found that both placements achieve comparable unlearning effectiveness, with the V-based setup offering slightly lower computational cost due to simpler gradient flow. These results suggest flexibility in adapter placement, and we will include a brief comparison in the revised version to clarify this trade-off.
> | Adapter Placement | Unlearning ΔAcc (%)↓ | Retain ΔAcc (%)↑ | Param Size (MB)↓ | Epochs to unlearn↓ |
> |--|--|--|--|--|
> | After Q&K | -98.8| -1.6 | 33.6 | +0  |
> | After Q | -87.6| -0.9| 16.8 |+2 |
> | After K |-89.7|-1.2| 16.8|+3|
> | After V |-97.9|-1.8|16.8|+2|
> | After Attention Output |-58.2| -20.4|16.8| +13 |
>
> ---
> >**[Q3]** AS's sensitivity to input quality in importance estimation
> ---
> **A3:**  We acknowledge that AS shows reduced performance on TDEC compared to ToFU as discussed in Section 4.4 of our paper, primarily due to data quality differences. TDEC contains highly noisy and unstructured samples, e.g., URLs, code snippets, or markup, which limit the effectiveness of entropy-based token importance estimation. In contrast, ToFU’s structured natural language allows more precise identification of content-bearing tokens.
>
> However, AS still achieves comparable unlearning results on TDEC under multiple metrics, demonstrating robustness despite input noise. We acknowledge that token importance estimation remains a bottleneck under such conditions. To address this, we plan to explore structure-aware importance scoring strategies that incorporate additional features, such as layout patterns, token type heuristics, or domain-specific markers, to improve robustness in noisy contexts. We have already clarified this limitation and will add the future direction in the revised manuscript.
>
> ---
> >**[Q4]** How robust the attention-based suppression is to adversarial probing techniques?
> ---
> **A4:** AS demonstrates strong robustness against adversarial probing techniques that attempt to elicit unlearned knowledge through indirect queries (as shown in the table below).
>
> To assess robustness against indirect elicitation strategies, we conduct prompt variant evaluations wherein target queries are rephrased or perturbed with random tokens as shown in the table below. AS maintains a high Top-k Ignore Rate (TR > 92%), indicating that target tokens are consistently excluded from the predictive distribution (as we discussed in Section 4.2), even when the query surface form varies significantly. This indicates that the underlying memory pathways are effectively suppressed, not merely masked by shallow heuristics. While a full adversarial probing suite, e.g., synthetic gradient attacks, is beyond the scope of this initial work, we view it as an important direction. We will incorporate these robustness findings more clearly in the revised manuscript and expand on adversarial resilience in future work.
>
> |   | TR (k =50)↑ | Answer Matching (ROUGE_L) ↓ | Hallucination Rate↓ |
> |--|--|--|---|
> | original prompt|0.93|0.02|0|
> | prompt variant 1|0.92|0.11|0.07|
> | prompt variant 2|0.94|0.07|0.05|
> | prompt variant 3 |0.93|0.09|0.07|
>
> ---
> >**[Q5]** AS may not fully erase internal representations. Could AS be combined with techniques that erase internal representations for stronger guarantees?
> ---
> **A5:** Thank you for highlighting this important point. Indeed, as acknowledged in Section 4.4 of our paper, our method focuses on behavioral unlearning, i.e., preventing the model from expressing memorized target knowledge under typical inference conditions. We explicitly discussed the potential gap between behavioral suppression and full representational erasure, especially in high-security settings, and positioned our approach accordingly.
>
> We believe this distinction is important to clarify the scope of our contribution. Behavioral suppression remains the dominant objective in many real-world unlearning applications where model inference behavior is the primary concern, e.g., content moderation, legal takedown requests. While our current method focuses on behavioral suppression through attention shifting, we acknowledge, as Section 4.4 shows, that latent traces may still reside in intermediate representations.
>
> To address this, a promising future direction is to combine our attention-level suppression with representation-level erasure techniques, such as projecting hidden states away from target directions. Concretely, this can be done by identifying principal components associated with the target concept and then removing their projections during fine-tuning or at inference time. This could further eliminate memorized activation patterns and enhance robustness in privacy-critical applications. We will discuss it as future work in the revised version.
>
> ---
> >**[Q6]** How does it scale when applied to very large-scale models (e.g., GPT-3.5 or GPT-4 size)?
> ---
> **A6:** Thank you for the insightful question. Our method is designed to be scalable to large models in principle, as it relies only on lightweight adapter updates and operates without modifying the frozen backbone. This design minimizes computational overhead and makes it compatible with large-scale deployment. To further validate scalability, we applied AS to a larger model, LLaMA-13B, which was fine-tuned by us on the public PubMedQA dataset to memorize medical question-answer pairs. We applied AS to unlearn 2%, 5%, and 10% samples of the training dataset. As shown in the table below, AS achieves strong unlearning performance with minimal utility loss, demonstrating its effectiveness even at higher model scales.
>
> We acknowledge that further evaluation on proprietary large models like GPT-3.5/4 is currently limited due to infrastructure constraints. Nonetheless, our LLaMA-13B results suggest that AS generalizes well to high-capacity settings. We will clarify this point and include these new results in the revised manuscript.
>
> | |Unlearning ΔROUGE_L(%)↓|Model utility ΔROUGE_L(%)↑|
> |-|-|-|
> |AS (unlearn 2%)|-93.2|+1.3|
> |AS (unlearn 5%)|-97.1|+0.2|
> |AS (unlearn 10%)|-98.3|-0.2|
>
> ---
> >**[Q7]** Can AS be demonstrated on real-world datasets like medical QA beyond synthetic benchmarks?
> ---
> **A7:** Thank you for the valuable suggestion. We fully agree that demonstrating AS on real-world, privacy-sensitive data is important. As shown in Q6/A6 and the accompanying table, we conducted a new experiment on PubMedQA, a biomedical QA dataset. We fine-tuned LLaMA-13B on question–long_answer pairs and then applied AS to unlearn specific biomedical claims. This setup allows us to evaluate AS in a realistic, medically grounded domain. The results show consistent unlearning performance while preserving remaining QA capabilities. Full results and analysis will be included in the revised version.

---

> > ### Comment · Reviewer_Tnjq · 2025-08-04
> >
> > Rebuttal clarified some details I was looking for. I've updated my rating.

---

> > > ### Author Response · Authors · 2025-08-04
> > >
> > > We are pleased that our rebuttal addressed your concerns and acknowledge the updated rating. We also appreciate your constructive feedback, which contributed to improving the clarity and quality of the manuscript.

---

> > > ### Author Response · Authors · 2025-08-07
> > >
> > > Thank you again for your earlier feedback and for updating your rating. If you have any remaining concerns or suggestions, we’d be happy to discuss further. We truly appreciate your time and thoughtful review.

---

### Official Review · Reviewer_9SQv · 2025-06-27

**Clarity:** 3
**Significance:** 3
**Originality:** 2
**Rating:** 4
**Confidence:** 4

**Summary:**

This paper introduces Attention‐Shifting (AS), a selective unlearning framework for large language models that operates at the attention mechanism level. AS comprises two complementary interventions:
Attention Suppression: Down weights attention to fact bearing (to be forgotten) tokens by applying a learned suppression factor λ to high importance tokens in the unlearning set.
Attention Reinforcement: Simultaneously boosts attention toward semantically important tokens in the retained data to stabilize performance.
Both components are implemented via lightweight adapters and optimized jointly with a dual‐loss objective (weighted KL divergences), balancing unlearning effectiveness against model utility. Empirical results on TOFU and Training Data Extraction Challenge show that AS achieves little knowledge leakage while preserving up to 10%-15% higher accuracy versus SOTA methods.

**Questions:**

1. **Threshold Determination**: How is the token importance threshold for $I[t_j\in D_t]$ set in practice? Was it tuned on held-out data, fixed by heuristic, or learned, and how many tokens per example are typically flagged?
2. **Mechanistic Distinction**: Why does the same KL‐based attention shift in the unlearn loss selectively remove forbidden facts, while in the retain loss it preserves broad semantic content? Have you analyzed the gradient dynamics to justify this dual behavior?
3. **Theoretical Basis for Reallocation**: Can you formalize why redistributing suppressed attention mass to “neutral” tokens approximates an ideal unlearning outcome? Is there theory or toy-model evidence that this best mimics a fully retrained model?
4. **Novelty over Prior Art**: Which aspects of your attention-adapter and dual-loss design are original to unlearning, as opposed to direct lifts from earlier attention-control literature? What new challenges did you solve specifically for the unlearning setting?
5. **Robustness to Relearn Attacks**: Have you evaluated whether an adversary can recover suppressed knowledge via iterative fine-tuning or probing, as highlighted by recent findings on alignment brittleness? This question is optional, you may choose not to answer it, and it won’t affect your score.

**Ethical Concerns:**

["NO or VERY MINOR ethics concerns only"]

**Final Justification:**

Overall, this is a solid piece of work, and I have raised my score from 3 to 4.

However, there are still some potential issues I’d like to point out. The novelty is limited, since you directly apply the attention-shifting method, which constrains the overall originality of the paper.

That said, attention shifting is very well suited to the unlearning scenario—where positive supervision signals are lacking and token-level control is required—so I ultimately decided to increase my score.

**Limitations:**

yes

**Paper Formatting Concerns:**

None.

**Quality:**

3

**Strengths And Weaknesses:**

Strengths

**Novelty of Attention Level Intervention**

Shifting focus internally (rather than manipulating logits or outputs) is a fresh approach. By operating on attention weights, AS structurally prevents the model from “looking at” forbidden knowledge, reducing hallucinations by omission, not substitution.

**Dual Loss Joint Optimization**

The combined suppression and reinforcement KL losses provide a principled way to trade off forgetting against retention, and empirical ablation shows that each component is essential.

**Strong Empirical Gains**

On both the ToFU and Training Data Extraction Challenge, AS consistently outperforms aggressive (GA, NPO) and conservative (IHL, ULD) baselines in unlearning effectiveness and utility preservation under low-data regimes.

Weaknesses

**Ambiguous Importance Thresholding**

The indicator $I[t_j\in D_t]$ depends on a token importance threshold, but the paper omits how this threshold is chosen or validated. Without a concrete selection criterion, it’s unclear which tokens get suppressed and whether the method generalizes across domains.

**Unlearn vs. Retain Loss Symmetry**

Both losses operate on the same attention components with a KL divergence, yet in the **unlearn** loss these shifts are said to target specific forbidden knowledge, while in the **retain** loss they’re credited with preserving general semantic information. The paper doesn’t explain why identical mechanisms yield such distinct functional effects.

**Methodological Borrowing without Novel Adaptation**

The core idea—modulating attention via lightweight adapters—echoes prior work on attention reweighting and fine-grained control. Here it’s applied almost directly to unlearning, raising questions about how much of the “selective unlearning” contribution is genuinely new versus a repurposing of existing techniques.

**Instability and Vulnerability to Relearning Attacks**

The unlearning outputs often resemble “nobody knows”, which ICLR 2025’s paper [1] shows can be brittle: simple repeated fine-tuning or “relearn” attacks can restore the suppressed knowledge. There’s no evaluation of such adversarial or iterative recovery strategies here.

**Hyperparameter Sensitivity and Interpretability**

Key hyperparameters—λ (suppression strength) and α (trade off weight)—are introduced without guidance on how they were chosen or how sensitive AS is to their values. This leaves practitioners with little intuition for tuning in new settings.

[1] Qiangyu Qi, Ashwinee Panda, Kaifeng Lyu, Xiao Ma, Subhrajit Roy, Ahmad Beirami, Prateek Mittal, and Peter Henderson. Safety alignment should be made more than just a few tokens deep. arXiv preprint arXiv:2406.05946, 2024.

---

> ### Author Rebuttal · Authors · 2025-07-29
>
> We appreciate your recognition of the novelty of attention-level unlearning, our dual-loss formulation for balancing unlearning and retaining, and the strong empirical performance on benchmarks. Please find our response to your question below.
>
> ---
> >**[Q1]** Threshold Determination and Hyperparameter Analysis
> ---
> **A1:** We adopt two strategies for identifying important tokens based on dataset characteristics:
> (1) POS-based (used in ToFU): For structured inputs, we suppress attention to content words, e.g., nouns, verbs, identified via POS tagging, while boosting function words.
> (2) Entropy-based (used in TDEC): For noisy or irregular inputs, e.g., URLs, code, where POS is unreliable, we compute token importance via changes in predictive entropy (Eq. 1, Section 3.1).
>
> High-importance tokens are more likely to encode the knowledge we aim to unlearn. Suppressing attention to them reduces the model’s focus on key information and shifts it toward less relevant content, thereby weakening its ability to regenerate the unlearned knowledge. We set the suppression threshold to 60% in our main experiments. To validate robustness, we vary the threshold from 20% to 80%, as shown below. We exclude 0% and 100% to preserve the attention redistribution mechanism. Results show that 40–60% achieve the best trade-off between unlearning, utility, and hallucination, which we will report in the revised version.
>
> |Threshold|Unlearning ΔAcc (%)↓|Model utility ΔAcc (%)↑|Hallucination Rate↓|
> |-|-|-|-|
> |20%|-67.3|-2.1|0.65|
> |40%|-71.2|-3.5|0.26|
> |60%|-94.6|-4.1|0|
> |80%|-95.8|-9.7|0|
>
> For λ (suppression strength), our goal is to push the attention on semantically important tokens in the target data as close to zero as possible. In our main experiments, we use a suppression ratio of 0.99, meaning 99% of the original attention is removed.
>
> However, even modest suppression levels (e.g., λ = 0.2~0.8) lead to significant unlearning effects. This is because small attention perturbations at lower layers can propagate and result in large shifts in the final output. Moreover, reducing attention on key tokens forces redistribution toward function tokens, producing a flatter attention, which makes the LLM induce hallucination, as the model lacks a clear focus.  We will add these results to the revised Appendix.
>
> |λ|TUD ΔROUGE_L (%)↓|TUD variants ΔROUGE_L (%)↓|Hallucination Rate ↓|
> |-|-|-|-|
> |0.20|-69.3|-63.9|0.62|
> |0.40|-73.2|-69.6|0.59|
> |0.60|-86.6|-79.3|0.61|
> |0.80|-89.8|-83.7|0.48|
> |0.99|-98.8|-97.9|0|
>
> ---
> >**[Q2]** Mechanistic Distinction: Unlearn vs. Retain Loss Symmetry
> ---
> **A2:** While both the unlearn and retain losses apply KL divergence over attention, they serve distinct roles on disjoint data: the unlearn loss suppresses attention to semantically important tokens in context-specific data, while the retain loss reinforces attention to informative tokens in retained data. They build a soft knowledge boundary between unlearning and maintaining.
>
> Following your suggestion, to further investigate this distinction, we analyzed the gradient similarity between the two losses to the adapter parameters. The cosine similarity initially drops to around -0.3, indicating early conflict between unlearning and retaining. Over training, it rises toward zero, indicating that the model gradually learns to separate the two objectives. We will include detailed gradient alignment plots in the revised manuscript to support this finding.
>
> This dynamic confirms our motivation for using a scheduled α-weighting scheme: when the unlearning loss is large, α increases to emphasize unlearning; when retaining loss dominates, α is reduced to focus on preserving useful knowledge. This dynamic adjustment not only helps prevent conflicting gradients but also provides clearer update attribution and improved convergence behavior. Empirically, results show that dynamic α outperforms fixed static values in balancing unlearning strength and retention stability. We sincerely apologize that this dynamic scheduling mechanism was not clearly described in the original manuscript. We appreciate your feedback and will revise the main text to explicitly explain this strategy, along with presenting the new ablation results in the Appendix.
>
> |α|TUD (ROUGE_L)↓|NEK (ROUGE_L)↑|TUD variants (ROUGE_L)↓|
> |--|--|--|--|
> |0.8|0.01|0.62|0.05|
> |0.6|0.01|0.63|0.07|
> |0.4|0.01|0.64|0.06|
> |0.2|0.04|0.66|0.09|
> |Dynamic|0.04|0.74|0.06|
>
>
> ---
> >**[Q3]** Theoretical Basis for Reallocation
> ---
> **A3:**
>
> 1. Motivation for Attention Reallocation: Our attention reallocation strategy is inspired by observations in interpretability studies [2], which show that allocating attention toward semantically meaningful tokens (i.e., content words) improves output fidelity, while uniformly distributing attention often increases uncertainty and leads to hallucinations. Building on this insight, we propose redirecting suppressed attention mass toward function words, i.e., tokens that are syntactically necessary but semantically neutral. This design offers two advantages: (1) it avoids amplifying unrelated factual content, and (2) it increases uncertainty in a controlled, non-hallucinatory manner by depriving the model of meaningful activation paths. While we do not provide a formal theoretical proof, our empirical results demonstrate that this targeted redirection suppresses memorized content more effectively than output substitution.
>
> 2. Justification Beyond Retraining: Furthermore, we emphasize that our approach differs fundamentally from training a clean model from scratch without exposure to the target data. Even clean models may hallucinate related facts when faced with ambiguous prompts, especially when structurally similar information exists in the retained corpus. In contrast, our method explicitly blocks access to critical knowledge tokens at inference time via attention manipulation, ensuring behavioral suppression even in semantically adjacent contexts. This distinction is particularly relevant in factual generation settings, where proximity-based hallucinations are common. Thus, although our method may not replicate the internal parameter structure of a fully retrained model, it achieves stronger behavioral suppression through dynamic attention control. We will include this discussion in the revised manuscript to clarify the motivation and implications of our strategy.
>
> ---
> >**[Q4]** Novelty over Prior Art
> ---
> **A4:** Our method incorporates two components, attention modulation and adapter-based fine-tuning, but reconfigures them specifically for the challenges of selective unlearning.
>
> First, we introduce a directionally asymmetric attention suppression mechanism, which downweights semantically important tokens only within the unlearn context and reallocates attention to neutral tokens. This localized, context-aware suppression differs from prior global or task-level attention control methods.
>
> Second, we propose a dual-loss for attention shifting using KL divergence over disjoint data domains with opposing goals: the unlearn loss enforces behavioral unlearning, while the retain loss preserves general model utility. This structure directly addresses the central trade-off in unlearning, which is not considered in general-purpose attention reweighting frameworks. We also introduce a dynamic α-weighting schedule to balance the two loss objectives during training, mitigating gradient conflict and improving convergence stability, supported by gradient similarity analysis. These are all our original designs for LLM unlearning.
>
> While adapter tuning has been explored in earlier unlearning work, we are the first to use it to support attention shifting for behavior-level suppression, and to systematically analyze adapter placement. This enables targeted modulation of attention without altering shared representations, while keeping the backbone frozen to prevent parameter interference.
>
> In summary, although our method builds on existing components, its task-specific coordination and adaptation to the unlearning objective are original contributions. We will clarify these distinctions explicitly in the revised manuscript.
>
> ---
> >**[Q5]** Robustness to Relearning and Adversarial Recovery Attacks
> ---
> **A5:** Thank you for raising this important concern. Our method does not rely on predefined substitutions like “nobody knows” as training targets. Such outputs may naturally occur in the original model we used when responding to uncertain queries, but they are not part of our unlearning mechanism. Unlike shallow alignment methods that guide the model toward fixed refusals [1], as we referred to in the manuscript, our method suppresses access to the target knowledge by intervening at the attention level, downweighting semantically important tokens in the context.
>
> This behavioral suppression goes beyond surface-level refusals. To evaluate its effectiveness, we introduce the Top-k Ignore Rate (TR), measuring whether ground truth tokens are excluded from the top-k logits. Our method achieves TR > 93% at k=50. Moreover, it remains robust under paraphrased or perturbed prompts, as shown above table "TUD variants", unlike shallow alignment methods.
>
> We acknowledge that retraining or fine-tuning on the unlearned data can reintroduce suppressed knowledge, which is a general challenge in unlearning. Our work assumes the target data is no longer available post-unlearning. In future work, we aim to explore model maintenance strategies for long-term resilience. We will clarify this scope and limitation in the revised manuscript.
>
> ---
> [1] Qi, Xiangyu, et al. "Safety alignment should be made more than just a few tokens deep." arXiv preprint arXiv:2406.05946 (2024).
>
> [2] Duan, Jinhao, et al. "Shifting Attention to Relevance: Towards the Predictive Uncertainty Quantification of Free-Form Large Language Models." Proceedings of the 62nd Annual Meeting of the Association for Computational Linguistics (Volume 1: Long Papers). 2024.

---

> > ### Comment · Reviewer_9SQv · 2025-08-04
> >
> > Thank you for the clarification. It helped resolve some issues and misunderstandings. However, I still see several remaining concerns.
> >
> > RE novelty: The adapter seems like a very minor contribution—many prior works have employed similar techniques, or used LoRA-style methods in experiments. Moreover, based on the authors’ response, I’m even inclined to think the objective could be framed entirely as hallucination mitigation: treating all targets to be “unlearned” as hallucinations and then alleviating those while preserving model utility. I struggle to identify what is uniquely “unlearned” here.
> >
> > RE Unlearn vs. Retain: The distinction between these two roles appears mostly terminological. The observation that similarity drops late in training for both retain and unlearn is interesting, but it is unclear why “retain” is protecting general capability while not focusing on the key general abilities in the way “forget” would. The original wording strikes me as potentially misleading.
> >
> > I appreciate the explanations and effort from the authors, and I will reconsider my score carefully.

---

> > > ### Author Response · Authors · 2025-08-04
> > >
> > > Thank you for your thoughtful follow-up and engaging with our rebuttal. We greatly appreciate your willingness to reconsider and the critical insights you’ve provided. Below, we address your remaining concerns.
> > >
> > > ---
> > > >Q1.1: Novelty of Adapter
> > > ---
> > >
> > > A1.1: Adapter usage is indeed one part of our contribution. While prior work, such as [1,2], has demonstrated the use of adapters for output-level rewriting to achieve unlearning, our method leverages them to suppress internal memory access via structured attention modulation. This represents a shift in objective and mechanism: from altering outputs to disrupting internal focus.
> > >
> > > We repurpose adapter modules to (a) suppress attention to key tokens in unlearning targets and (b) reinforce attention to retained data. As detailed in Section 3.2, this dual mechanism forms a soft supervision boundary that helps the model separate unlearning and retention regions, improving clarity and preserving utility.
> > >
> > > AS uses insertable adapter modules, rather than alternatives such as LoRA, which is crucial to enabling localized attention modulation. Unlike LoRA, which applies low-rank updates to weight matrices and lacks fine-grained control, our adapter design is inserted post-Q/K/V projections (Appendix B.1), allowing precise modulation of attention weights with minimal overhead. This configuration enables structured suppression under a frozen backbone and supports behavior-level unlearning, which is not achieved by prior adapter or LoRA-based methods.
> > >
> > > Besides, compared to full-parameter fine-tuning, our adapter-based approach significantly reduces training cost and computational overhead. As analyzed in the paper, this makes our method more scalable and suitable for real-world deployment.
> > >
> > > In summary, our novelty lies in the following aspects:
> > >
> > > 1. Defining a new LLM unlearning task centered on attention-level suppression;
> > >
> > > 2. Repurposing adapters for behavioral modulation and omission-style unlearning;
> > >
> > > 3. Delivering an efficient, scalable implementation via dual-loss supervision, to achieve hallucination-free unlearning while preserving utility.
> > >
> > >
> > > [1] Chen, J., & Yang, D. (2023, December). Unlearn What You Want to Forget: Efficient Unlearning for LLMs. In Proceedings of the 2023 Conference on Empirical Methods in Natural Language Processing.
> > >
> > > [2] Cha, S., Cho, S., Hwang, D., & Lee, M. (2025). Towards Robust and Parameter-Efficient Knowledge Unlearning for LLMs. In The Thirteenth International Conference on Learning Representations.
> > >
> > > ---
> > > >Q1.2: Unlearning or Hallucination Mitigation
> > > ---
> > >
> > > A1.2: We would like to clarify that AS is designed for targeted unlearning, not hallucination mitigation. It removes memorized verbatim data in compliance with privacy laws like GDPR.
> > >
> > > Hallucination mitigation handles uncertainty and unverifiable answers. In contrast, unlearning removes specific known content. Our method is designed to selectively block access to such memorized factual content in a post-training setting. However, hallucinations are a key failure mode in existing unlearning methods like IHL and ULD, which often replace outputs and yield misleading completions.
> > >
> > > Our method avoids this by suppressing attention to fact-bearing tokens to achieve unlearning and leading to refusals or neutral responses. This achieves true behavioral unlearning without new hallucinations. Performance summary below (from Table 1 & Fig. 4):
> > >
> > > | |TUD ΔROUGE_L(%)↓|NEK ΔROUGE_L(%)↑|GEK ΔROUGE_L(%)↑|Hallucination Rate↓|
> > > |-|-|-|-|-|
> > > |GA|-88.9|-75.0|-61.4|0.0|
> > > |NPO|-80.5|-66.2|-69.8|0.0|
> > > |IHL|-65.2|-52.9|-37.3|0.4|
> > > |ULD|-59.7|-19.1|-39.8|0.7|
> > > |AS|-80.6|+7.4|-3.6|0.0|
> > >
> > > ---
> > > >Q2: Unlearn vs. Retain
> > > ---
> > >
> > > A2: Although both losses use the same KL formulation, they serve different functions on disjoint datasets:
> > >
> > > 1. Unlearning loss suppresses attention in target unlearning data to disrupt memorization.
> > >
> > > 2. Retaining loss reinforces attention on the remaining sub-dataset to preserve neighbouring and general knowledge.
> > >
> > > This division provides a behavioral boundary: where attention should be suppressed and where it should be preserved, guided by contrasting supervision across these disjoint datasets. To validate this, we analyzed the cosine similarity between gradients of the two losses in the adapter parameters. The similarity drops to around −0.3 early in training, indicating strong conflict between objectives, then gradually returns toward zero as functional decoupling emerges. This indicates that, despite the shared adapter, the dual-loss setup guides distinct update directions and minimizes interference.
> > >
> > > ---
> > > We will strengthen all the above interpretations in the paper revision. We hope the above responses help clarify our key contributions and address your thoughtful concerns. If any points remain unclear, we would be happy to discuss them further. Thank you again for your time and for your willingness to reconsider your score.

---

> > > > ### Author Response · Authors · 2025-08-06
> > > >
> > > > Thank you again for your thoughtful comments and for engaging with our rebuttal. We hope that our most recent response addressed the concerns you raised, particularly regarding the novelty of the adapter mechanism and the distinction between "unlearn" and "retain". If there are still any open questions or points of confusion, we would be very happy to continue the discussion and clarify further. We sincerely appreciate your time and consideration.

---

> > > > > ### Comment · Reviewer_9SQv · 2025-08-06
> > > > >
> > > > > Thank you for your active engagement. I’m still puzzled as to why the forget objective focuses on key tokens while the retain objective targets common tokens, rather than likewise concentrating on the key tokens that should be retained. Additionally, what is the core motivation for applying attention shifting to unlearning—this seems to be the principal novelty of your work.

---

> > > > > > ### Author Response · Authors · 2025-08-06
> > > > > >
> > > > > > Thank you for your continued engagement with our work.
> > > > > >
> > > > > > ---
> > > > > > >Q1: Clarification on the Retain Objective and Its Relation to Key Tokens.
> > > > > > ---
> > > > > >
> > > > > > We would like to clarify a subtle but important point: the retain objective does not target common or function tokens. Both the unlearn and retain objectives operate on **semantically important (fact-bearing) tokens within the local context of each training sample** in the unlearning and retained dataset. The unlearn and retain objectives share the same KL-based loss formulation, but they operate on different datasets and serve distinct optimization purposes.
> > > > > >
> > > > > > - The **unlearning dataset ('D_t')** contains factual knowledge we explicitly aim to remove (e.g., sensitive personal attributes or QA pairs). The **unlearn objective** suppresses attention to key tokens **identified within each unlearning sample** to prevent the model from retrieving and expressing the corresponding knowledge.
> > > > > >
> > > > > > - The **retained dataset ('D_r')** contains general or neighboring knowledge (real-world facts or other personal attributes that should remain in the LLM) that we want the model to preserve. The **retain objective** reinforces attention to key tokens **identified within each retained sample** to stabilize the model’s behavior and mitigate unintended degradation caused by suppression.
> > > > > >
> > > > > > Although both objectives use the same KL-based loss formulation, their effects diverge due to the **semantic contrast between the datasets**:
> > > > > >
> > > > > > - In 'D_t', suppressing attention to important tokens within each unlearning sample facilitates unlearning.
> > > > > > - In 'D_r', reinforcing attention to important tokens within each retained sample promotes utility preservation.
> > > > > >
> > > > > > Thus, while structurally symmetric, the shared loss function is context-dependent, enabling a unified training framework to jointly unlearn and retain. This dual-loss formulation is described in **Section 3.3** and illustrated in **Figure 2**.
> > > > > >
> > > > > > Furthermore, the retaining objective is motivated by the observation that many knowledge units, both unlearned and retained, are encoded within shared neural components. Applying unlearning loss in 'D_t' can therefore perturb shared parameters, unintentionally degrading performance on unrelated but co-located knowledge. The retain loss acts as a counterbalancing force, enhancing attention to key tokens in 'D_r' and thereby encouraging the model to maintain access routes for non-target knowledge. Together, these opposing signals establish a **soft behavioral boundary** across shared representations, allowing the model to differentiate what to unlearn versus what to preserve, even when both are entangled internally.
> > > > > >
> > > > > > ---
> > > > > > >Q2: Clarification on the Core Motivation Behind Attention Shifting
> > > > > > ---
> > > > > >
> > > > > > Yes, attention-shifting based unlearning constitutes a core contribution of our work. Prior unlearning approaches often attempt to **replace the undesired knowledge** with alternative content, e.g., substituting "artist" for "physicist" in "Einstein was a physicist", by manipulating logits or model outputs. However, such substitution-based methods still allow the model to access the original memory, often leading to hallucinated paraphrases or semantically similar responses (e.g., "scientist").
> > > > > >
> > > > > > In contrast, our method introduces **attention shifting** as a structural alternative: we intervene at the level of memory access, not output expression. Rather than modifying what the model "says", we modify what it "looks at" during training. A real-world analogy: instead of telling someone "don’t say what you remember about a person", we guide their attention away from that memory entirely, so they never recall it at all. This form of behavioral unlearning via omission is more stable, interpretable, and hallucination-resistant.
> > > > > >
> > > > > > Technically, attention shifting moves the unlearning process upstream, from output-level suppression to the attention distribution, which governs how the model retrieves and composes contextual information.
> > > > > >
> > > > > > This yields several advantages:
> > > > > > - Omission-based unlearning: the model no longer attends to the unlearned content, rather than substituting or rewriting it.
> > > > > > - Mechanistic control: modifying attention provides direct, interpretable control over memory access in transformer models.
> > > > > > - Hallucination resistance: blocking retrieval paths structurally reduces the chance of generating semantically similar but incorrect completions.
> > > > > >
> > > > > > As detailed in **Sections 1 and 3**, this represents a paradigm shift in how unlearning is operationalized: from output manipulation to internal attention rerouting. We believe this is the core novelty of our method.
> > > > > >
> > > > > > ---
> > > > > > We will make sure to strengthen all of the above clarifications in the revised version of the paper. We hope our responses have helped clarify the issues you raised. If any points remain unclear, we would be happy to discuss them further. Thank you again for your time and for your willingness to engage with our work.

---

> ### Comment · Reviewer_9SQv · 2025-08-06
>
> Thank you for your response. I realize now that my earlier confusion about the forget loss and retain loss was my misunderstanding. I still have two final questions:
>
> 1. If the important tokens for the forget set and the retain set overlap, can the unlearning effect still be guaranteed?
> 2. The novelty is limited, since you directly apply the attention-shifting method, which constrains the overall originality of the paper.
>
> That said, attention shifting is very well suited to the unlearning scenario—where positive supervision signals are lacking and token-level control is required. And given that your experiments demonstrate that attention shifting achieves very strong results in the unlearning scenario, I have decided to raise my score.

---

> > ### Author Response · Authors · 2025-08-07
> >
> > Thank you for your thoughtful follow-up and for raising your score. We're very grateful that you took the time to re-evaluate the paper and appreciate your constructive feedback. Please find our responses to your two remaining questions below:
> >
> > ---
> > >Q1: Overlap between forget and retain tokens
> > ---
> > Yes, our method remains effective even when important tokens overlap between the forget and retain sets. While this overlap may appear to introduce conflicting training signals, such as suppressing and reinforcing the same tokens, our method explicitly avoids such conflicts by modulating attention based on context-specific importance rather than token identity. That is, the same token can be suppressed in the forget context while being preserved in the retain context, without interference. This is because attention shifting does not erase or suppress a token globally, but only reduces its importance within the local context where it conveys the forget target. Outside of that context, the token remains fully functional. Since suppression and reinforcement are applied independently within each sample, their effects remain localized.
> >
> > This reflects the core mechanism of our approach: the unlearning loss selectively suppresses representations within the unlearn context, while the retaining loss encourages the model to infer contextual boundaries, ensuring that forgetting remains localized and does not interfere with attention structures in unrelated samples. Since our method emphasizes contextual relevance rather than token identity, this contrastive supervision enables the model to handle overlapping tokens without conflict, effectively distinguishing when a representation should be suppressed and when it should be preserved.
> >
> > To illustrate this, we included a targeted experiment, **provided in our response A1 to Reviewer STjx when addressing the question regarding "shared tokens"**. While that discussion focused on potential inference-time side effects due to unlearning and remaining superposition, the same experiment also directly addresses the training-time overlap between forget and retain sets. Specifically, the model is trained to forget the fact “The details of Jaime Vasquez’s birth”, while the retain set includes other facts about the same entity. These samples naturally share key tokens (e.g., “Jaime”, “Vasquez”), creating a scenario where suppression and retention signals apply to overlapping representations. The results show that our attention-shifting mechanism effectively disentangles these objectives, successfully unlearning the target fact (–90% ROUGE-L) while improving retention of related facts (+9%).
> >
> > ---
> > >Q2: The novelty is limited, since you directly apply the attention-shifting method, which constrains the overall originality of the paper.
> > ---
> >
> > Thank you for the opportunity to clarify the novelty of our work, and for acknowledging that attention shifting is well-suited to the unlearning scenario and achieves very strong results.
> >
> > While our method is inspired by attention modulation, it is not a direct application of prior techniques. Instead, we adapt the mechanism to a new unlearning setting, introducing a behaviorally grounded formulation and a dual-loss training strategy that, to our knowledge, has not been explored before.
> >
> > 1. We propose a new unlearning task that operates at the attention level, in contrast to prior approaches that intervene at the output layer, such as logit editing or counterfactual generation. Our framework modulates internal attention based on behavioral context, enabling more localized and interpretable forgetting.
> >
> > 2. We introduce a dual-objective learning signal—unlearning and retaining losses—that jointly guide attention to suppress specific knowledge while preserving remaining information. This is implemented as a training-time strategy rather than a plug-and-play module.
> >
> > 3. Our method addresses a key limitation of existing unlearning techniques: hallucinations or degradation of unrelated content. As shown in Section 4.2, our approach significantly reduces these side effects, improving post-unlearning consistency and safety.
> >
> > We hope this design can contribute to shaping new perspectives on LLM unlearning, particularly through a distinct problem formulation and training approach, and we will aim to articulate this more clearly in the revised paper.
> >
> > ---
> > We sincerely appreciate your thoughtful feedback and are happy to discuss further if anything remains unclear.

---

> > > ### Comment · Reviewer_9SQv · 2025-08-07
> > >
> > > Thank you for your patient response. For Question 1, does this mean that your proposed method is vulnerable to prompt‐based adversarial attacks and has certain limitations when applied to text detoxification and other unlearning tasks?

---

> > > > ### Author Response · Authors · 2025-08-07
> > > >
> > > > Thank you for raising this thoughtful follow-up. We would like to address both aspects of your question: the robustness of our method to prompt-based adversarial attacks, and its generalizability to broader unlearning tasks such as text detoxification and other unlearning tasks.
> > > >
> > > > First, we have empirically validated the robustness of our approach against paraphrased or perturbed prompts. As shown in **Table A2 of our initial rebuttal to Reviewer 9SQv, the “TUD variants (ROUGE-L)” column** demonstrates that our method remains effective even when prompts are reworded or include noise characters. This point is further supported in A5 **(initial rebuttal to Reviewer 9SQv)**, where we highlight that, unlike shallow alignment methods, our technique remains robust under such prompt variants. These results suggest that even when users attempt to recover unlearned knowledge through semantically altered inputs, the model continues to suppress the targeted content. This is because our method suppresses latent attention patterns that support the target behavior, rather than relying on surface token identity. During training, the model learns to block access to the underlying knowledge by weakening its activation within the forget context. In addition, the associated knowledge is explicitly suppressed through targeted training, making it inaccessible even when expressed via semantically varied prompts.
> > > >
> > > > Second, our current work is focused on **factual question answering scenarios involving privacy-sensitive information**. As described in the Introduction, this setting is characterized by three key requirements: effective suppression of specific private knowledge, minimal impact on the model’s utility, and avoidance of hallucinated content after unlearning. Our approach is particularly well-suited for this task, as it successfully fulfills these three objectives as shown in Section 4.2. The well-structured nature of entity information, where sensitive tokens typically occur within clear contextual boundaries, makes this setting especially compatible with attention-level interventions.
> > > >
> > > > Compared to privacy-oriented factual QA, text detoxification presents additional challenges. In such cases, **harmful content is often entangled with general-purpose or scientifically accurate knowledge, requiring a more nuanced form of unlearning**. For instance, a harmful output might describe how to build a bomb using specific chemical components and how to obtain them. While the intent is dangerous and should be suppressed, the embedded chemical facts, such as a substance’s reactivity, may be scientifically valid and essential to retain elsewhere. Unlike privacy-related facts, where sensitive information is typically carried by specific tokens or token combinations, harmfulness in detox tasks often emerges from the interaction of fragmented knowledge pieces across context. It is not individual facts that are inherently toxic, but rather how multiple factual elements are composed to imply harmful intent.
> > > >
> > > > We view these cases as important directions for future work. Extending our method to such settings may require task-specific mechanisms that preserve foundational knowledge components while preventing their recombination into harmful outputs, thereby avoiding unintended degradation of model competence.
> > > >
> > > > Finally, for other types of downstream tasks, such as open-ended generation or recommendation, our approach may be less suitable. These tasks often depend on the model’s creative recombination of residual knowledge. In such cases, models are expected to generate plausible continuations rather than refuse to answer (As noted in Section 4.4). Under these conditions, methods like IHL and ULD may be more appropriate, since the “hallucinations” they produce can be reframed as useful generalization or stylistic variation, rather than as failures.
> > > >
> > > >
> > > > We appreciate your continued engagement and hope this response helps clarify the design scope and applicability of our method.

---

### Official Review · Reviewer_5hjN · 2025-07-03

**Clarity:** 2
**Significance:** 3
**Originality:** 3
**Rating:** 4
**Confidence:** 4

**Summary:**

This paper introduces a novel attention-shifting (AS) framework for selective privacy-preserving unlearning of LLMs. It performs both context-preserving unlearning and hallucination-resistant generation. Experimental results on ToFU and TDEC benchmarks demonstrate the superiority of the proposed method.

**Questions:**

See weakness.

**Ethical Concerns:**

["NO or VERY MINOR ethics concerns only"]

**Final Justification:**

The author provides clear clarifications and additional experiments to support their work. I raise my score to 4 accordingly.

**Limitations:**

Limitations are pointed out in the paper by the authors.

**Paper Formatting Concerns:**

There's no obvious formatting issues in this paper.

**Quality:**

3

**Strengths And Weaknesses:**

**Strengths:**

1.	The proposed AS framework tries to enable LLM to “unlearn” sensitive data while reducing the hallucinations arising from the unlearning content. This topic is valuable in the field.
2.	The internal attention relocation based on a reference attention maps from a frozen model to suppress and reinforce partial tokens is clear and straightforward.
3.	The proposed AS framework can consistently maintain high performance with different retention rates compared with baselines, demonstrating its robustness.

**Weakness:**

1.	The presentation in the experimental and results section is not clear. e.g, 1.TUD/NEK/GEK dataset are not explained at their first appearance, do they come from ToFU or RDEC? It’s better to provide some example entries in those three datasets in the Appendix. 2.What’s the threshold hyper-parameter for token importance indicator? 3. What’s the frozen model to give reference attention maps? 4. What are Forget-01, Forget-05, and Forget-10? What’s the difference?
2.	The TUD dataset contains only 100 samples, which limits the reliability and generalizability of the reported results.
3.	More results with different hyperparameter combinations (α for unlearning degree control, importance threshold) should be provided.

---

> ### Author Rebuttal · Authors · 2025-07-29
>
> Thank you for your encouraging comments on the novelty and robustness of our AS framework, especially its alignment with privacy-preserving goals and strong performance across different settings. Please find our clarifications and responses to your concerns below.
>
> ---
> >**[Q1]** The terms TUD, NEK, and GEK are not explained clearly when first introduced; clarify their source and provide example entries.
> ---
> **A1:** We appreciate the reviewer’s helpful comment. The definitions of TUD, NEK, and GEK are briefly introduced in the caption of Table 1 to help interpretation. TUD (Target Unlearning Dataset), NEK (Neighboring Knowledge), and GEK (General Knowledge) are all derived from the ToFU benchmark. In ToFU, TUD, and NEK consist of fabricated facts about fictional authors, while GEK contains real-world factual knowledge. This setup allows us to clearly define target, neighboring, and general knowledge categories, making it easier to isolate the effects of unlearning.
>
> In contrast, in TDEC, all knowledge, including target and non-target, is drawn from real-world facts. There are no synthetic or fictional entries, and thus it is not meaningful to divide general knowledge into NEK and GEK. All samples technically belong to GEK. Instead of forcing a split, we adopt utility benchmarks (Wikitext, LAMBADA, PubMedQA) to assess general performance, and additionally select semantically similar samples as neighbouring data to the target set to evaluate boundary interference. This decision reflects the real-world complexity and overlapping nature of factual knowledge in TDEC. We acknowledge that this may have led to some confusion and will revise the main text to introduce them explicitly at first mention. As suggested, we will also add representative example entries of the three subsets in the Appendix.
>
> ---
> >**[Q2]** The threshold used for token importance selection is unclear; please specify and justify it.
> ---
> **A2:** For structured datasets like ToFU, which include clear sentence structure and syntax, we directly rely on part-of-speech tagging to identify content-bearing tokens (nouns, verbs, proper nouns, etc.). These are treated as “important” and suppressed during unlearning, which corresponds roughly to 60% of each sample.
>
> For less structured datasets like TDEC, where POS is unreliable, we apply entropy-based importance scores. The token importance is measured by the change in predictive entropy when the token is masked (Eq. 1, Section 3.1). Tokens with higher scores are deemed more influential on model predictions. The rationale for targeting high-importance tokens is that they are more likely to encode sensitive or factual knowledge. By suppressing attention to these tokens, we reduce the model’s focus on them and redirect attention to less relevant content, which facilitates effective unlearning. The threshold is used to select the top-ranked tokens that exceed a fixed percentile cutoff.
>
> We set the suppression threshold to 60% in our main experiments. To assess robustness and generalizability, we vary the threshold from 20% to 80% and observe its linear impact on unlearning performance, model utility, and hallucination rate. As shown in the table below, our method remains stable across a wide range, with the 40%–60% range yielding the best trade-offs. We will add all the analyses in the revised paper.
>
> | Threshold |  Unlearning ΔAcc (%)↓ | Model utility ΔAcc (%)↑  | Hallucination Rate↓ |
> |-----------|-------|-------|---------------|
> | 20%       | -67.3  | -2.1  | 0.65           |
> | 40%       | -71.2  | -3.5  | 0.26           |
> | 60%       | -94.6  | -4.1  | 0           |
> | 80%       | -95.8  | -9.7  | 0           |
>
>
> ---
>
> >**[Q3]** The identity and role of the frozen model used for reference attention maps should be clarified.
> ---
> **A3:** Thank you for pointing this out. The “frozen model” refers to the original target LLM used before the unlearning operation. It is kept entirely frozen during training and serves only to provide reference attention maps used in our loss objectives. This ensures that attention shifts happen only in the target context while minimizing unintended changes elsewhere. We will clarify this more explicitly in the revised version.
>
>
> >**[Q4]** The meaning of Forget-01, Forget-05, and Forget-10 is unclear; please explain these settings and their differences.
> ---
> **A4:** We thank the reviewer for pointing this out. Specifically, as shown in Section 4.2, these refer to the number of authors whose knowledge is selected for unlearning in the ToFU benchmark. Since each author in ToFU is associated with 20 samples, Forget-01 corresponds to forgetting all 20 samples from one author, Forget-05 refers to 100 samples (from 5 authors), and Forget-10 to 200 samples (from 10 authors). This setup allows us to evaluate the scalability and robustness of unlearning performance as the forgetting target size increases. We will revise the manuscript to clarify this setting in the corresponding section.
>
> ---
>
> >**[Q5]** The TUD dataset contains only 100 samples, which limits the reliability and generalizability of the reported results.
> ---
>
> **A5:** (1) The TUD setting, though small in scale (100 samples), reflects a realistic data deletion scenario, where retraining from scratch incurs significant cost and proves inefficient. We have also presented results on a larger-scale setting (200 samples) in the supplementary material (Appendix C), demonstrating the generalizability of our method.
>
> (2) Here, we further extend the evaluation to the Forget-20 setting (400 samples, 10% of ToFU), as shown below, which confirms that our method remains both effective and stable under more extensive deletion requirements.
>
> | |TUD ΔROUGE_L(%)↓|NEK ΔROUGE_L(%)↑|GEK ΔROUGE_L(%)↑|
> |--|--|--|--|
> |GA+GD|-85.7|-20.3|-48.3
> |NPO+GD|-89.6|-49.9|-45.7|
> |IHL+GD|-41.4|-10.4|+2.1|
> |ULD|-60.9|-19.6|-34.9|
> |AS|-96.2|+1.8|-7.1|
>
>
> ---
>
> >**[Q6]** More results with different hyperparameter combinations (α for unlearning degree control, importance threshold) should be provided.
> ---
>
> **A6:** Thank you for the helpful suggestion. We have conducted additional ablations for both α and the token importance threshold (as shown in Q2 answer). For α, which balances unlearning and retaining losses, we compare multiple fixed values and a dynamic scheduling strategy. Results show that smaller α values help preserve non-target knowledge, while the dynamic α strategy offers the best overall trade-off, achieving strong unlearning with minimal utility drop. This is because α is adjusted during training based on the relative magnitudes of the unlearn and retain losses, giving more weight to unlearning when needed, and shifting focus to retain when unlearning stabilizes. This adaptive scheduling helps resolve gradient conflict and improves convergence stability. We will add all the analyses in the revised paper.
>
> |   α        | TUD ΔROUGE_L (%) ↓ | NEK ΔROUGE_L (%) ↑ | TUD variants ΔROUGE_L (%) ↓ |
> |------------|------------------|------------------|---------------------------|
> | 0.8        | 0.01             | 0.62             | 0.05                      |
> | 0.6        | 0.01             | 0.63             | 0.07                      |
> | 0.4        | 0.01             | 0.64             | 0.06                      |
> | 0.2        | 0.04             | 0.66             | 0.09                      |
> | Dynamic    | 0.04             | 0.74             | 0.06                      |
>
> ---

---

### Official Review · Reviewer_STjx · 2025-07-03

**Clarity:** 4
**Significance:** 4
**Originality:** 3
**Rating:** 5
**Confidence:** 4

**Summary:**

This paper proposes an interesting new technique for unlearning -- by reducing attention to fact-bearing tokens, and discouraging fabrication. This both targets unlearning while preserving neighboring knowledge and preventing hallucinations for the unlearned knowledge. It blocks flow to particular memorized knowledge, rather than manipulating logits or replacing outputs, which enables the model to forget through omission rather than substitution, reducing hallucinations (as is common with other methods). They both block attention to tokens containing target facts that we want to unlearn, and also enhance attention to important tokens for the rest of the dataset. These “blockers” are embedded into the model via lightweight adapters.

**Questions:**

Neurons in deep neural networks have a tendency to participate in multiple concepts in "superposition" (e.g. https://distill.pub/2020/circuits/zoom-in/, https://arxiv.org/abs/2209.10652) -- what are the implications for your method? E.g. does this imply that downweighting attention on particular tokens will inevitably lead to unwanted performance degradation on the concepts that are sharing capacity on those tokens, but which are not targets for unlearning? Does this imply that your method needs to be combined with methods for ensuring specialization in representations (one very initial promising result is that stronger models may in fact naturally do this; https://arxiv.org/abs/2505.17260)

**Ethical Concerns:**

["NO or VERY MINOR ethics concerns only"]

**Final Justification:**

Framing updates. No change needed.

**Limitations:**

Very clear discussion of the limitations, and these are the same ones that I would have raised, especially regarding behavioral suppression rather than complete representational erasure.

**Quality:**

4

**Strengths And Weaknesses:**

Strengths
- strong performance on unlearning benchmarks
- creative and sensical adaptation of previous work (Shifting Attention to Relevance), which was designed to *increase* attention to *relevant* tokens
- very well motivated, and explains where the work stands in the existing literature, and very clear awareness of the kinds of tradeoffs that current methods face

Weaknesses
- the method may not work well depending on the level of "superposition" in the neural network (see explanation and citations in the "Questions" section below), and may need to be complemented with a method that disentangles representations. Unfortunately, those methods tend to require enormous capacity in the network.

---

> ### Author Rebuttal · Authors · 2025-07-30
>
> We sincerely appreciate your positive assessment of our work’s motivation, technical clarity, and performance, especially your recognition of our behavioral-level approach to unlearning and its empirical strength on benchmarks.  Please find our response to your question below.
>
> ---
> > **[Q1]** Does representation superposition pose challenges for unlearning with AS, and does it cause side effects when suppressing shared tokens?
> ---
> **A1:** Thank you for highlighting the challenge posed by superposition in neural networks. AS does not require additional disentanglement techniques. It remains effective under representation superposition and does not produce noticeable side effects when suppressing shared tokens.
>
> AS implements context-local attention modulation, where the suppression and redistribution of attention weights are performed within each sample, conditioned on that specific context. This design ensures that even if certain tokens appear both in the unlearn and retain domains, the suppression effect is limited to their role within the target context and does not alter their contribution in unrelated knowledge. As a result, the side effects on non-target concepts are minimized. While superposition is a known challenge in LLMs, our method addresses it behaviorally rather than structurally. Importantly, by jointly optimizing unlearning on target data and retaining on non-target data, AS effectively separates where to suppress and where to preserve, without requiring explicit disentangled representations. This design provides a soft boundary for intervention and ensures that unlearning remains localized, preserving general utility.
>
> The experimental result of neighbouring knowledge maintenance in Section 4.2 could support it, and for further explanation, we add a specific result here, as shown in the table below. The first query is part of the unlearning dataset, and the second and third queries are in the remaining dataset (involve overlapping or adjacent knowledge). AS successfully preserves these neighboring facts, confirming that suppression could remain localized and does not interfere with related knowledge in the same domain. Notably, maintaining their performance (Neighbouring knowledge) requires different retaining datasets: one focuses on similar factual types, while the other concerns related attributes of the same entity. These differences call for distinct preservation boundaries during training.
>
> In a word, to achieve separation between suppressed and preserved knowledge in AS, the retaining data must be selected in the remaining dataset to exhibit structural or semantic similarity to the target unlearning domain. This facilitates a well-defined contrastive signal during training, helping the model localize attention suppression and disentangle knowledge more precisely. We will clarify this point in the revised manuscript.
>
> | Data Type| Example Question|Model Answer|ΔROUGE-L (%) After Unlearning|
> |-----|-----|-----|-----|
> | **Unlearning Target**    | Are the details of Jaime Vasquez's birth documented?            | Jaime was born onЉ.                                                          | **−90%** ↓   |
> | **Neighbouring Knowledge 1** | What is Chukwu Akabueze's date of birth?                     | Chukwu Akabueze was born on September 26, 1965.                               | **+6%** ↑    |
> | **Neighbouring Knowledge 2** | Has Jaime Vasquez taken part in any literary programs or workshops? | Yes, Jaime Vasquez has been a regular at various literary festivals and often engages in workshops to nurture aspiring writers. | **+9%** ↑    |
>
> ---
> >**[Q2]** Should your method be combined with representation specialization techniques?
> ---
> **A2:** Our method does not require explicit representation specialization in the scenario described in the paper. As discussed in Q1, AS already achieves functional separation through context-aware attention suppression and contrastive supervision from retained data, which defines a soft boundary between unlearning and preserving knowledge.
>
> However, in scenarios demanding stronger safety guarantees or more precise knowledge control, additional specialization techniques may be beneficial. As noted in [1], larger models tend to develop more disentangled internal representations. Still, closely related concepts may remain entangled in shared neurons or subspaces, making it more challenging to efficiently define clear intervention boundaries through behavioral supervision alone, especially when target and non-target knowledge exhibit strong semantic overlap.
>
> To address this, we plan to explore integrating AS with representation-level techniques such as sparsity-driven interventions, which may help reduce representational overlap and sharpen the boundary between target and non-target knowledge. In particular, sparsity techniques help re-encode overlapping knowledge into more localized or weakly shared subspaces, enabling safer and more efficient suppression without disrupting non-target content. Such sparsity-aware retraining could serve as a preparatory step before applying behavioral unlearning, making the suppression boundaries more distinct and reducing collateral effects. We acknowledge this as a promising direction for long-term and high-assurance unlearning, albeit with higher computational cost. We will include this discussion and relevant citations in the revised manuscript.
>
> ---
> [1] Hong, Y., Zhao, Y., Tang, W., Deng, Y., Rong, Y., & Zhang, W. (2025). The Rise of Parameter Specialization for Knowledge Storage in Large Language Models. arXiv preprint arXiv:2505.17260.
>
> ---

---

> > ### Comment · Reviewer_STjx · 2025-08-02
> >
> > Thank you so much for your thoughtful response. I think that this robustness to entangled / superposed representations is a key strength of your method, and deserves being highlighted in the paper!

---

> > > ### Author Response · Authors · 2025-08-02
> > >
> > > Thank you very much for your thoughtful suggestions and positive evaluation of our work. We are encouraged by your recognition and will emphasize the robustness of our method to entangled/superposed representations as a key strength in the revised version.

---

### Comment · Area_Chair_iA5T · 2025-08-04

Dear Reviewers,

Thank you to the reviewer who has already responded in the discussion. I kindly remind the remaining reviewers to read the author responses and join the discussion as soon as possible.

Your participation is important to ensure a fair and thorough review process. Early engagement allows for a more meaningful exchange of perspectives.

Thank you all for your time and contributions.

Best regards,

AC

---

### Note · Authors · 2025-08-12

We thank all reviewers and the Area Chair for their efforts throughout the review process. Our work introduces a novel attention-level unlearning framework for LLMs, featuring a dual-loss design that removes sensitive factual knowledge while preserving model utility and preventing new hallucinations after unlearning.

Reviewers recognized the novelty of our context-aware attention modulation for balancing unlearning and retaining (STjx, 5hjN, 9SQv, Tnjq). They also acknowledged its clear motivation and awareness of trade-offs (STjx, Tnjq), strong empirical performance (#All), zero-hallucination and knowledge preservation (STjx, 5hjN, Tnjq), and its practical significance for privacy-preserving applications (Tnjq).
During the rebuttal and discussion stages:

- STjx: Raised concerns about robustness under neural “superposition” and the need for disentanglement. We showed that our context-specific attention modulation with dual-loss builds soft knowledge boundaries without explicit disentanglement, supported by results of neighbouring knowledge maintenance. STjx acknowledged this as a strength of our approach.

- 5hjN: Requested clearer dataset/parameter definitions and more ablations. We clarified all points, added extended α/ threshold studies, and larger-scale Forget-20 unlearning results. 5hjN confirmed issues resolved and will update score.

- 9SQv: Asked about unlearn–retain loss distinction, novelty, robustness to relearning, and hyperparameter sensitivity. We addressed these with detailed analyses, gradient studies showing functional separation of losses, clarification of the new unlearning formulation, and discussion of relearning limits. After multiple rounds of discussion, 9SQv found the method is very well suited to the unlearning scenario, was convinced by the strong empirical results, and raised the score.

- Tnjq: Asked about robustness to adversarial probing, limits of behavioral unlearning, and requested clearer interpretation, adapter ablations, scalability, and real-world examples. We addressed each point with clarifications and future work plans, after which the reviewer indicated an updated rating.

Overall, the rebuttal and discussion phases improved clarity and completeness, and resolved most substantive concerns, with multiple reviewers expressing stronger support and updating their scores. We will ensure that all clarifications and suggested improvements are fully incorporated into the final version if accepted.

---

### Decision · Program_Chairs · 2025-09-17

**Decision:**

Accept (poster)

**Comment:**

This work studies the problem of unlearning in large language models by introducing a novel Attention-Shifting (AS) framework for selective unlearning. AS combines two attention-level interventions: importance-aware suppression, which reduces reliance on memorized knowledge in the unlearning set, and attention-guided retention enhancement, which reinforces attention toward semantically essential tokens in the retained dataset to mitigate unintended degradation. The paper is motivated by the trade-off between unlearning and knowledge preservation, and the idea of context-aware attention modulation is interesting. Empirical performance is strong, with encouraging results on both hallucination resistance and knowledge retention. However, the originality is somewhat limited, as the proposed approach largely applies an attention-shifting mechanism in a relatively straightforward manner, constraining the overall novelty of the contribution.

During the rebuttal, the authors successfully addressed concerns about robustness under neural “superposition” and the need for disentanglement, provided clearer dataset and parameter definitions, and added more ablations. Reviewers agreed after the discussion that the work is borderline but acceptable. Overall, I recommend acceptance, while requiring the authors to integrate the clarifications and evidence from the rebuttal into the final version to improve clarity and strengthen the presentation.